# Efficacy of *Saccharomyces cerevisiae* Fermentation Product and Probiotic Supplementation on Growth Performance, Gut Microflora and Immunity of Broiler Chickens

**DOI:** 10.3390/ani14060866

**Published:** 2024-03-12

**Authors:** Stephen Soren, Guru P. Mandal, Samiran Mondal, Saktipada Pradhan, Joydip Mukherjee, Dipak Banerjee, Manik C. Pakhira, Anjan Mondal, Victor Nsereko, Indranil Samanta

**Affiliations:** 1Department of Animal Nutrition, West Bengal University of Animal and Fishery Sciences, Belgachia, Kolkata 700037, India; drsoren1507@gmail.com (S.S.); gpmandal1@gmail.com (G.P.M.); 2Department of Veterinary Pathology, West Bengal University of Animal and Fishery Sciences, Belgachia, Kolkata 700037, India; vetsamiran@gmail.com (S.M.); saktipadapradhan@gmail.com (S.P.); 3Department of Veterinary Physiology, West Bengal University of Animal and Fishery Sciences, Belgachia, Kolkata 700037, India; joyphy@gmail.com (J.M.); dipakndri@gmail.com (D.B.); 4Department of Avian Sciences, West Bengal University of Animal and Fishery Sciences, Mohanpur, Nadia 741252, India; drpakhira@yahoo.co.in; 5Department of Veterinary Microbiology, West Bengal University of Animal and Fishery Sciences, Belgachia, Kolkata 700037, India; anewviewaj18@gmail.com; 6USAID-TRANSFORM, Cargill Inc., Bengaluru, Karnataka 560103, India; anjan_mondal@cargill.com; 7USAID-TRANSFORM, Cargill Inc., Minneapolis, MN 55440-9300, USA; victor_nsereko@cargill.com

**Keywords:** *Bacillus subtilis*, broiler, fermentation product, gut microflora, *Saccharomyces cerevisiae*

## Abstract

**Simple Summary:**

The use of non-therapeutic antibiotics in poultry industry is a common practice to generate desired growth of the birds. Non-therapeutic uses of antibiotics were detected as a culprit for the generation of antibiotic-resistant bacteria in the food chain. Many countries restrict the use of non-therapeutic antibiotics. Consequently, there is an increased demand for alternatives to antibiotic growth promoters (AGPs). The study was conducted to assess the impact of dietary supplementation of a probiotic and a postbiotic (*Saccharomyces cerevisiae* fermentation product, SCFP) on growth performance, carcass traits, blood haemato-biochemical profile, gut microflora, gut morphology, and immune response in broilers as an alternative to AGP. The study conducted on 324 one-day-old chickens revealed significant improvement of feed conversion ratio in the postbiotic group than the control. Cholesterol levels and concentrations of corticosterone were significantly lowered in the postbiotic group compared to other groups. Pathogenic- and antibiotic-resistant Gram-negative bacterial populations were significantly lower in postbiotic and probiotic groups. Moreover, humoral immunity was significantly improved in postbiotic and probiotic groups than the control birds. It can be concluded that both postbiotics and probiotics could be viable alternatives to antimicrobials in poultry production.

**Abstract:**

Concern for global health security and the environment due to the emergence of antibiotic-resistant bacteria and antibiotic residues in meat and other livestock products has led many countries to restrict the use of antibiotics in animal feed. This experiment was performed to assess the impact of dietary supplementation of a probiotic (*Bacillus subtilis*) and a postbiotic (*Saccharomyces cerevisiae* fermentation product) on growth performance, carcass traits, blood haemato-biochemical profile, gut microflora, gut morphology, and immune response in broilers as an alternative to antimicrobials in poultry production system to minimize the effect on global health security. A total of 324 one-day-old Ven Cobb 400 broiler chicks were randomly divided into three dietary groups, each containing 12 replicated pens, and each replicate contained nine chickens. The dietary groups consisted of (1) a basal diet without any growth promoters (T_1_), (2) the basal diet augmented with *Bacillus subtilis* at 200 g/MT feed (T_2_), and (3) the basal diet supplemented with *Saccharomyces cerevisiae* fermentation product at 1.25 kg/MT feed (T_3_). To calculate body weight gain, all birds and residual feed were weighed on a replicated basis on days 0, 7, 14, 21, 28, 35, and 42; mortality was recorded daily. At the end of the trial (42 d), two chickens from each replicate were slaughtered for carcass traits, gut microflora, and morphology measurements. Blood samples were collected for the haemato-biochemical profile on 35 d and antibody titer on 28 d and 35 d. Feeding with SCFP (T_3_ group) significantly improved average daily feed intake (ADFI) and average daily gain (ADG) of chickens compared to the T_1_ (control) and T_2_ (probiotic) groups from 1 to 14 days of age. Feed conversion ratio (FCR) was significantly improved in SCFP-fed birds (T_3_) relative to the control (T_1_) over the entire experimental period. Carcass traits and blood haemato-biochemical parameters remained unaffected by any diets. However, cholesterol levels and concentrations of corticosterone were significantly lower in T_3_ compared to T_2_ and T_1_ groups. Total *E. coli*, Enterohaemorrhagic *E. coli*, ESBL-producing *Enterobacteriaceae*, and *Salmonella* counts were significantly lower in T_2_ and T_3_ groups compared to T_1_ group and *Salmonella* counts were lower in T_3_ when compared to T_2_. However, there was no significant difference in *Lactobacillus* count among treatment groups. A significant increase in villi height and villi-height-to-crypt-depth ratio (VH: CD) was observed in both T_3_ and T_2_ groups. On day 28, the T_3_ and T_2_ groups exhibited a significant increase in antibody titers against Newcastle disease virus and infectious bursal disease virus. It can be concluded that *Saccharomyces cerevisiae* fermentation product and *Bacillus subtilis* probiotic could be viable alternatives to antimicrobials in poultry production considering beneficial impacts in broilers fed an antibiotic-free diet.

## 1. Introduction

One of the most profitable and productive agricultural industries is the poultry industry. Recent advancements in nutrition, genetics, housing management, chicken health and welfare have allowed it to flourish, resulting in a potential growth of egg production of 8.51% and broiler production of 7.52% [1]. The use of antibiotics as growth promoters has resulted in high levels of poultry output worldwide; these antibiotics have impacted chickens’ intestinal flora and immune systems to aid in controlling infections [2,3]. Concern for global health security and the environment due to the emergence of antibiotic-resistant bacteria and antibiotic residues in meat and other livestock products has led many countries to restrict the use of antibiotics in animal feed [4]. This has encouraged nutritionists and feed manufacturers around the world to search for alternatives to antibiotic growth promoter (AGPs) that can maintain efficient poultry production while ensuring that poultry meat and eggs are safe for consumption. Possible replacements for AGPs include feeding prebiotics, probiotics, synbiotics, enzymes, herbs, essential oils, acidifying feed with organic acids and postbiotics [5].

While probiotics have many positive health benefits, their functionality and effectiveness are subject to debate. Recent findings suggest that for a variety of animal species, probiotics need to be tailored more specifically in order to maximize their beneficial effects. Furthermore, certain strains of probiotic bacteria were discovered to have antibiotic-resistant genes which can be transmitted to gut microflora and the environment [6,7]. Additionally, studies showed that some probiotics can have a detrimental effect on the host by causing local inflammation in healthy hosts and exacerbating tissue inflammation in those with inflammatory bowel disease [8]. The ‘postbiotic’ has emerged which extends the scope of the probiotic concept beyond its inherent viability [9]. The term ’postbiotic’ refers to the soluble factors (stabilized bacteria, cellular products, or metabolic by-products) secreted by living microbes or released after microbial lysis [10], which are mainly derived from *Lactobacillus*, *Bifidobacterium*, *Streptococcus*, fecal bacteria [11,12], and *Saccharomyces cerevisiae* yeast [13,14]. Recent research suggests that postbiotics offer various health benefits through immune system modulation (cell wall compounds may strengthen immunity), increased adhesion to intestinal cells (which restricts pathogen growth), and secretion of various metabolites [11,15]. Non-viable micro-organisms or microbial cell extracts have an additional advantage over probiotic-supplemented feed preparations, as the viability of probiotics may differ and dead cells may outnumber the live cells [16]. Moreover, these non-viable microbes and extracts can significantly reduce shelf life of the poultry products [17]. The present study was conducted to detect the effects of *Bacillus subtilis* as a probiotic and *Saccharomyces cerevisiae* fermentation product (SCFP) as a postbiotic on the growth performance, carcass characteristics, gut microflora and immunity of broiler chickens as an alternative to antimicrobials in poultry production system to minimize the effect on global health security.

## 2. Materials and Methods

### 2.1. Birds, Diets and Experiment

Mixed-sex one-day-old broiler chickens (Vencobb 400, Venkys, Pune, India, n = 324) were randomly divided into three groups based on experimental diets containing 12 replicated pens and nine chickens in each replicate. 

The dietary groups consisted of (1) a basal diet without any growth promoter (T_1_); (2) the basal diet plus the probiotic *Bacillus subtilis* (Zeus Biotech Pvt Ltd, Mysore, India) at the rate of 200 g/MT feed (T_2_) (3) the basal diet plus the postbiotic *Saccharomyces cerevisiae* fermentation product (SCFP; Diamond V Original XPC) at the rate of 1.25 kg/MT feed (T_3_). The basal diet based on maize–soybean in mash form was formulated to meet or exceed nutritional requirements of broiler starter (day 1–14), grower (day 15–28) and finisher (day 29–42) chickens using a ration formulation software as per the commercial Vencobb 400 broiler chicken recommendations [18]. The basal diet contained 22.19% of crude protein (CP) with 3000 kcal/kg of metabolizable energy (ME) for broiler starters (day 1–14), 20.80% of CP with 3100 kcal/kg of ME for broiler growers (day 15–28), and 19.2% of CP with 3200 kcal/kg of ME for broiler finishers (day 29–39) (Table 1 and Table 2). Premixes of probiotic and postbiotic were arranged separately before addition into the basal diet to obtain experimental probiotic and postbiotic diets in a feed mixer. The process was repeated each week to produce fresh feed. The experimental feeds were stored in high density polyethylene bags containing an inner liner. *Ad libitum* diets and water were provided to the experimental birds. The chickens were purchased from Institutional Animal Ethics Committee (IAEC)-approved commercial vendors.

### 2.2. Supervision and Rearing Conditions of Birds

Before arrival of the birds the experimental farm, feeding and watering troughs were properly disinfected. Birds were kept in floor pens (1.22 m × 0.76 m) and the pens were separated by plastic were netting. Sterile plastic feeding and watering troughs were provided in each pen. Litter was prepared with chopped paddy straw and rice husk. Compressed fluorescent lamps were used for continuous lighting during the first two days of brooding and later it was modified to generate 23 h of light interrupted by one hour of darkness.The experimental poultry house temperature was controlled by the heating elements throughout the experiment. The temperatures were gradually decreased from 32 °C on day 1 to 24 °C on day 22. Proper ventilation was ensured through the use of exhaust fans during the entire trial period. Vaccination against Newcastle disease virus (NDV) and infectious bursal disease virus (IBDV) was conducted on all the birds at 5 and 21 days of age and at 12 days of age, respectively [19]. 

### 2.3. Performance Traits of Birds

On day one, body weight (BW) of all the chickens was measured, followed by weekly assessments and one final measurement taken in the morning on the last day of the growth trial. Average BW was calculated for each replicate. Weekly feed intake was determined by subtracting remaining feed from total feed offered per pen. Total feed intake per day in each pen was divided by the total number of chickens present to calculate average daily feed intake (ADFI). Cumulative feed intake and weight gain in each pen was considered to calculate feed conversion ratio (FCR; grams of feed intake per gram of growth). Death of the birds in each pen was monitored regularly to calculate mortality rate, if any, and a post-mortem examination was conducted to know the cause. At the end of the trial, mortality percentage for each replicate was calculated and used to adjust BW, ADFI and FCR calculations following the method described earlier [20]. 

### 2.4. Detection of Carcass Traits in Slaughtered Birds

To evaluate carcass traits, two birds from both the sexes having an average BW close to the average BW of the pen (replicate) were selected for slaughter by cervical disarticulation. After removal of skin, feather, head, shank, intestine and giblets, the eviscerated carcass weight was measured and the weight of different organs was detected separately.

### 2.5. Serum Biochemical Analyses from Collected Blood Samples

Blood samples for hematobiochemical and hormone assay were collected at day 35 after 12 h of fasting. Two aliquots of blood samples were collected from wing vein of broiler chickens (2 birds were randomly chosen from each pen, total 24 birds per treatment). The first aliquot was placed in a tube containing anticoagulant (EDTA) for measurement of hemoglobin (Hb), total leukocyte counts (WBCs), and differential leukocyte counts, heterophil:lymphocyte (H:L) ratio at day 35 as per standard hematological procedure [21]. The second aliquot of blood samples was collected without anticoagulant and serum was harvested and stored at −20 °C until analysis. Concentrations of serum metabolites (glucose, total protein, albumin, uric acid, triglycerides, cholesterol) were measured using commercial kits (DiaSys diagnostic India Pvt. Ltd., Mumbai, India). Concentrations of corticosterone were determined by commercial ELISA kit (DRG Diagnostics, Marburg, Germany).

### 2.6. Pre-Cecal Bacterial Count

The pre-cecal contents of the intestine were aseptically collected from the chickens after the slaughter at day 42 and were placed into sterile sample collection bag (HiMedia, Mumbai, India). The samples were processed on the same day for bacteriological study. Population of total bacteria, pathogenic/zoonotic *Escherichia coli* (Enterohaemorrhagic *E. coli*), antimicrobial resistant bacteria (*ESBL*-producing *Enterobacteriaceae*), *Salmonella,* and *Lactobacillus* in pre-cecal content were determined as per standard procedure [22]. Pre-cecal content (1 gm) was serially 10-fold diluted with sterile phosphate buffer saline (PBS), 10 μL was placed on sorbitol-MacConkey agar (for Enterohaemorrhagic *E. coli*, HiMedia, India), ESBL agar (for ESBL-producing *Enterobacteriaceae* (KESC group), HiMedia, India), xylose lysine deoxycholate agar (for *Salmonella*, HiMedia, India), *Lactobacillus* agar (HiMedia, India) and were incubated at 37 °C for 24 or 48 h, and the characteristic colonies for each bacterial population were enumerated in a digital colony counter (HiMedia, India) and the numbers were expressed as Log10 colony-forming units (CFUs) per gram of sample. The selected isolates (n = 15, 5 isolates from each treatment group) of ESBL-producing *Enterobacteriaceae* (KESC group) were further characterized with PCR for presence of ESBL/beta-lactamase variants (*bla_CTM-Type_, bla_SHV-Type_, bla_TEM-Type_*) [23]. 

### 2.7. Histopatyhology of Small Intestine

In total, 24 chickens from each dietary treatment (on day 42) were selected for collection of small intestinal tissue to measure the height and width of the intestinal villus and crypt depth. After removal of the small intestine, the sections (2–3 cm) of jejunum (between the entry of bile duct and Mackel’s diverticulum), duodenum and ileum were rinsed with sterile PBS. One centimeter cross-sections of the tissues were fixed in buffered formaldehyde solution (100 mL/L; pH 7.2), which was followed by paraffin wax embedding. Delafield’s hematoxyline and eosin was used for staining of the tissue sections. Following the standard histopathological method, slides were mounted on distreneplasticiser xylene (DPX) [24]. An ocular micrometer (under a microscope fixed with stage micrometer) and image analysis software (Biowizard 4.2, Dewinter Optical and New Delhi, India) were used to detect all the measurements. Presence of intact lamina propria was used as a criterion for villus selection (12 villi per section). The height (μm) from the tip of the villus to the villus–crypt junction was considered as villus height (VH) and depth of the invagination between two villi was considered as crypt depth (CD). Three sections with 10 observations were conducted for each sample and mean of the values (μm) was used to generate a single observation. 

### 2.8. Detection of Antibody Titre

Measuring the antibody titer against Newcastle disease virus (NDV) and infectious bursal disease (IBDV) was considered as a parameter for detection of humoral immunity. Live lentogenic B1 strain for NDV (eye drop, 0.2 mL; Venkateshwaara Hatcheries Pvt Ltd, Pune, India) and live lentogenic LaSota strain for NDV (eye drop, 0.2 mL; Venkateshwaara Hatcheries Pvt Ltd, India) were administered on day 5 and 21, respectively. IBD live intermediate plus type (eye drop, 0.2 mL; Venkateshwaara Hatcheries Pvt Ltd, India) vaccine was administered on day 14. On day 28 and 35, whole blood (2 mL) was collected from the wing vein of two birds randomly selected from each replicate pen and the serum was collected by centrifuging the whole blood. 

The NDV and IBDV antibody titers were detected by an ELISA kit (IDEXX Laboratories Inc., Westbrook, ME, USA). Mean optical density (OD) value was calculated from the OD value recorded for each of the samples in duplicates. 

Cell-mediated immunity was measured by in vitro phagocytic activity of neutrophils and lymphocyte proliferation response at day 35 [25].

### 2.9. Chemical Analysis of Feeds Samples

Dry matter (DM; method 934.01), crude fiber (CF; Foss Fiber Cap 2021 Fiber Analysis System, Foss Analytical, Hilleroed, Denmark), calcium content, CP (method 968.06; Kelplus, Pelican Equipments, Chennai, India), ether extract (EE; method 920.39; Socsplus, Pelican Equipments, Chennai, India) of the feed samples and AIA content of the diets were analyzed following the methods described earlier [26,27,28]. 

### 2.10. Statistical Analysis

One-way analysis of variance (ANOVA) using SPSS [29] was used to analyze the data in a randomized design containing pen as an experimental unit for feed intake, FCR, and body weight gain, each individual bird as experimental unit for other parameters and treatment as the main effect. The homogeneity criteria were confirmed in mortality data, hence not included for statistical analyses. Probability values of *p* ≤ 0.01 were declared as trend and *p* ≤ 0.05 were declared as significant. The differences among the treatment means were detected using Tukey’s test when the treatment effect was significant.

## 3. Results

### 3.1. Average Daily Gain, Feed Intake and Feed Efficiency

The average daily gain (ADG) of the birds increased more significantly (*p* < 0.05) in the T_3_ group than the T_1_ and T_2_ groups from 1 to 14 days of age (Table 2). No significant differences among treatment groups were found in the rest of the experimental period or over the entire experiment period (1–42 d). The average daily feed intake (ADFI) of chickens was significantly higher (*p* < 0.05) in T_3_ groups compared with that of the T_1_ and T_2_ groups from 1 to 14 days of age. Subsequently, ADFI was significantly lower in both T_2_ and T_3_ treatment groups in comparison to the control (T_1_) from 15–28 days of age. However, there were no significant differences between treatment groups from 29–42 days or over the entire experiment period (1–42 d). No significant differences in FCR were noted during the starter (1–14 days) and finisher (29–42 days) phase. During the grower phase (15–28 days), FCR tended to be lower (*p* < 0.07) in SCFP-fed birds (T_3_) compared to T_1_ and T_2_. The FCR during the overall period (1–42 days) for the T_3_ group was improved (*p* < 0.05) in comparison to the T_1_, while the T_2_ group was not different from T_1_ and T_3_. 

### 3.2. Carcass Traits

No statistically significant differences (*p* > 0.05) were observed in slaughter BW, eviscerated carcass weight, dressing percentage, breast, frame, thigh, drumstick, wing, neck, gizzard, liver, heart, spleen, bursa and abdominal fat weight in grams across the various treatment groups (Table 3).

### 3.3. Blood Biochemical Profile

The concentration of glucose, total protein, albumin, triglyceride and uric acid in serum did not vary significantly (*p* > 0.05) based on dietary treatments in this study (Table 4). However, corticosterone and cholesterol concentration was reduced significantly (*p* < 0.05) in the SCFP-fed group (T_3_) compared to probiotic (T_2_) and control (T_1_) groups. 

### 3.4. Blood Hematological Profile

The blood hematological profile of broilers in different experimental groups has been presented in Table 5. No statistically significant differences (*p* > 0.05) were observed in hemoglobin, total leukocyte count (TLC), difference leukocyte count-heterophil, eosinophil, basophil, lymphocyte, monocyte and ratio of heterophil and lymphocyte across the various treatment groups.

### 3.5. Gut Microflora

Dietary treatments did not have a notable impact on the counts of *Lactobacillus spp*. in the pre-cecal digesta (Table 6). The count of *Enterohaemorrhagic E. coli* and total *E. coli* were significantly higher (*p* < 0.05) in the T_1_ group compared to the T_2_ and T_3_ groups. Additionally, the count of ESBL-producing *Enterobacteriaceae* (KESC group) and *Salmonella* were significantly higher (*p* < 0.01) in the T_1_ group when compared to T_2_ and T_3_ groups. *Salmonella* was also lower in T_2_ when compared to T_1_ (*p* < 0.05). PCR based characterization of ESBL- producing *Enterobacteriaceae* revealed maximum presence of *bla_CTX-M-Type_* (6/15, 40%), followed by *bla_SHV-Type_* (5/15, 33.3%) and *bla_TEM-Type_* (4/15, 26.6%). No variation in the possession pattern of ESBL/beta-lactamase associated genes in the studied ESBL-producing *Enterobacteriaceae* isolates was observed between the treatment groups. 

### 3.6. Gut Morphology

The VH in the duodenum was significantly increased (*p* < 0.05) in the T_3_ and T_2_ groups compared to the T_1_ group (Table 7). The VH in the jejunum was significantly higher (*p* < 0.05) in the T_3_ and T_2_ groups than T_1_ group. The VH in the ileum was higher (*p* < 0.05) in the T_3_ group than the T_2_ and T_1_ groups. The VW in the duodenum, jejunum and ileum was similar among the treatments (*p* > 0.05). The CD in the duodenum was increased (*p* < 0.05) in T_1_ group compared to the T_3_ and T_2_ groups. The CD in the jejunum was higher (*p* < 0.05) in the T_1_ group than T_3_ and T_2_ groups. The CD of the ileum was increased (*p* < 0.05) in the T_1_ and T_2_ groups compared to T_3_. The VH/CD ratio in the duodenum and jejunum was higher (*p* < 0.05) in the T_3_ and T_2_ groups than T_1_. The VH/CD ratio in the ileum was higher (*p* < 0.05) in the T_3_ group than T_1_ and T_2_ groups.

### 3.7. Immune Response

On day 28, antibody titers against the IBD vaccine were significantly higher (*p* < 0.05) in the T_3_ groups compared to the T_2_ and T_1_ groups (Table 8). However, on Day 35, there were no significant differences (*p* > 0.05) among the dietary treatment groups in antibody titers against this vaccine. Moreover, on day 28 antibody titers against the ND vaccine were also greater in T_2_ and T_3_ groups (*p* < 0.05) than the T_1_ group. In addition, no significant differences were detected between the dietary treatment groups on day 35 for this vaccine (*p* > 0.05). There were no significant differences among the treatment groups for in vitro phagocytic activity of neutrophils and lymphocytes.

## 4. Discussion

The solution-based approach to increase poultry production, to reduce production cost and to decrease negative environmental impact is the priority for poultry researchers. Modern poultry production systems are associated with numerous stressors, such as change of feed, high stocking density and processing in the hatchery, which reduce bird immunity and increase bacterial pathogen colonization—affecting not only bird health and growth, but also compromising food safety [30]. Use of antibiotics in sub-therapeutic doses in poultry feed was considered as one approach to control gut pathogens. Currently, non-therapeutic use of antibiotics in poultry is facing reduced social acceptance as it may generate antimicrobial-resistant commensals compromising food safety and quality. The European Union, and the United States FDA, banned the non-therapeutic use of antibiotics in livestock and poultry long ago [31,32], but cessation of non-therapeutic antibiotic usage in poultry farming was correlated with reduced growth and increased mortality of the birds due to bacterial infections such as colibacillosis, salmonellosis and necrotic enteritis [33]. Replacement of antibiotics with a suitable alternative without hampering the growth, immunity and health of the birds is a pressing research question. *Saccharomyces cerevisiae* is considered the most promising candidate either as a probiotic (live yeast form) or as prebiotic in the poultry diet which showed remarkable improvement of growth performance, modulation of bird immune system, repairing the gastrointestinal tract and reducing the gut pathogen colonization [34]. So, the present study was conducted to evaluate the effects of postbiotic (*Saccharomyces cerevisiae* fermentation product, SCFP) along with a probiotic (*Bacillus subtilis*) on the growth performance, immunity, gut health, and carcass characteristics of broiler chickens.

Feeding with SCFP (T_3_ group) significantly improved average daily feed intake (ADFI) and average daily gain (ADG) of chickens compared to the T_1_ (control) and T_2_ (probiotic) groups from 1 to 14 days of age. Similarly, feeding with yeast hydrolysate significantly improved ADFI, ADG, and body weight during the starter and grower phase of the experimental birds compared to the control groups [34,35]. It could be explained with the increased villi height associated with better absorption of nutrients, increasing the secretion of auxiliary digestive enzymes and anti-inflammatory effects of yeast hydrolysate in animals [36,37]. In contrast, a few studies [38,39] reported improvement of body weight gain during the later phase (after 21 days) of the growth with the feeding of yeast hydrolysate, associated with presence of gut microbiota-secreting short chain fatty acids (SCFAs) and improved metabolic activities. Although not evaluated, the findings of present study could be correlated with the presence of SCFA-forming beneficial gut microbiota during the starter and grower phase of the growth. Significant improvement of FCR in SCFP-fed birds (T_3_), compared to the control (T_1_) groups across the entire period of the experiment (1–42 days), is supported with the earlier findings [38,39]. The meta-analysis of the findings [40] suggested inclusion of yeast or yeast products (less than 10 g/kg of diet) could improve growth and FCR of the birds. 

The absence of statistically significant differences in slaughter body weight, eviscerated carcass weight, dressing percentage, weight of breast, frame, thigh, drumstick, wing, neck, gizzard, liver, heart, spleen, and bursa between the treatment groups is corroborative with the earlier studies [35,41]. Addition of probiotics in the diet helps in the detoxification process, which might be the reason for the normal size of the liver in the treatment groups [42].

Dietary addition of SCFP in the experimental birds did not alter the concentration of glucose, total protein, albumin, triglyceride and uric acid in serum, which confirmed the absence of adverse side effects in the studied birds [35]. In agreement with earlier reports [43,44], the present study also confirmed significant reduction of blood cholesterol concentration in SCFP-treated birds compared to the control or probiotic-fed groups. Lower serum concentration of cholesterol in the birds is associated with production of eggs with a low cholesterol level, which is especially popular among health-conscious consumers [45]. Although the present study was conducted in broilers, it can be conducted in layers in future to observe the production of eggs with low cholesterol. During stress, the hypothalamus pituitary–adrenal axis secretes corticosterone as the major hormone, which can depress humoral immunity and decrease production of antibodies against sheep red blood cells [46]. Reduced level of corticosterone in T3 group of birds during day 28 of the experiment can be correlated with increased antibody titer against NDV and IBDV. Further, corticosterone was found to be associated with upregulation of CCLi2 mRNA expression in splenic lymphocytes which attract active lymphocytes from the peripheral blood to the spleen. Corticosterone-associated upregulation of CXCLi1 and CXCLi2 mRNA expression in peripheral lymphocytes instead attracts heterophiles from bone marrow which are mostly immature [47]. The replacement of matured lymphocytes with immature heterophiles in the peripheral blood circulation was found to be responsible for decreased phagocytic activity. In the present study, a reduced corticosterone level in the T3 and T2 groups in comparison to T1 was found to be associated with increased (non-significant) phagocytic activity of the lymphocytes.

The present study revealed that dietary supplementation of SCFP had no significant effect on hemoglobin, total leukocyte count, difference leukocyte count and ratio of heterophil and lymphocyte, which was also observed in a previous study in which dietary supplementation of *Saccharomyces cerevisiae* with *Nigella sativa* did not find any significant effect on blood biochemical profile in broiler chickens [48]. 

The effect of SCFP dietary supplementation on poultry gut microflora revealed a significant reduction of total *E. coli*, pathogenic *E. coli* (EHEC) and *Salmonella* in comparison to the probiotic-fed group and control birds. Reduction of *E. coli* and *Salmonella* colonization was also observed in earlier studies in the birds fed with the yeast products which could be explained by exclusion of the pathogens due to competition for a carbon source in the gut, binding of the pathogens with a surface of yeast-produced functional carbohydrates instead of intestinal receptors—which prevent activation of pro-inflammatory cytokines-based signaling pathways—and production of enzymes to disintegrate bacterial toxins [30,49]. *Saccharomyces cerevisiae* was found to be more effective against Gram-negative pathogens such as *E. coli* and *Salmonella* due to its capacity to disintegrate the bacterial outer membrane—which is found only in Gram-negative bacteria, causing increased permeability and depolarization of the cytoplasmic membrane [50]. Agglutination of pathogens expressing mannose specific type-1 fimbriae (such as *E. coli* and *Salmonella*) by the yeasts is another possible mechanism [51]. 

Dietary treatments did not have a notable impact on the counts of *Lactobacillus* in the pre-cecal digesta. Similarly, feeding with dried yeast culture [48] and other prebiotics [52] did not reveal significant modulation on *Lactobacillus* count in broiler chickens. *Lactobacillus* itself can act as a probiotic by preventing colonization of gut pathogens and the lactic acid produced by the lactobacilli is used by butyric acid producers, increasing the digestibility of the birds [53]. Hence, in the present study, maintenance of lactobacilli in the treatment groups, compared to the control group, seems to be beneficial. 

One of the noteworthy findings of the present study is significant reduction of antimicrobial-resistant pathogens (ESBL-producing *Enterobacteriaceae*) in the treatment groups in comparison to the control group. *Bacillus subtilis* probiotic strains earlier showed in vitro antimicrobial effect against ESBL-producing *E. coli*, although failed to prevent gut colonization of ESBL-bacteria when studied in vivo [54]. There is no report on the efficacy of SCFP on ESBL-producing *Enterobacteriaceae* to compare the present finding. The present study revealed maximum occurrence of *bla_CTM-Type_* followed by *bla_SHV-Type_* and *bla_TEM-Type_* in the studied birds which is supportive of earlier studies. The CTX-M is considered as the major ESBL determinant in apparently healthy poultry, whereas the SHV and TEM determinants are predominant in poultry with subclinical infections [55].

The villi height in the duodenum, jejunum and ileum was significantly increased in the birds supplemented with SCFP and probiotic compared to the control group, which confirms the earlier observations [34,56]. In addition, the ratio between villi height and crypt depth was significantly increased in the ileum of SCFP-fed group compared to the birds supplemented with probiotic and the control group. *Saccharomyces cerevisiae* has a trophic effect on ileal and jejunal villi compared to the duodenum as detected in the present study, which is consistent with earlier observations [57]. The ileum is the primary site for amino acid absorption and longer ileal villi implies higher nutritional utilization reflected in better growth performance. 

On day 28, antibody titers against both the IBD and NDV vaccine were significantly higher in the SCFP (T_3_) group compared to the probiotic (T_2_) and control (T_1_) groups. The oligosaccharides present in the yeast hydrolysate can activate the macrophages and the cytokines are released to generate the acquired immune response [35]. As with mammals, the immune response in birds after vaccination is characterized with the generation of IgM first (up to day 30 post-vaccination) followed by IgY [58]. The previous study explored dietary supplementation of yeast products to promote the production of IgM in the birds vaccinated against NDV [39], which is the reason for the higher antibody titer in the T3 group compared to the others on day 28. The effect of yeast supplementation on the generation of IgY is still unclear and it might explain the absence of variations in all the groups in antibody titer on day 35. However, significantly higher antibody titer against IBDV in the birds fed with SCFP was not detected earlier as the earlier studies with SCFP focused on NDV only. In India, both the ND and IBD are considered as major viral infections, producing constraints in optimum production [59] for which the study objective took an inclusive approach to consider both. 

The present study could not find modulation of cell-mediated immune response in the studied birds, which was more pronounced in challenge studies—especially with intracellular pathogens (for example, *Coccidia*) fed with yeast hydrolysate and was also dependent on the dosage of the yeast products [60]. 

## 5. Conclusions

Feeding with SCFP significantly improved average daily feed intake and average daily gain of chickens compared to the control and probiotic groups from 1 to 14 days of age. FCR was significantly improved in SCFP-fed birds relative to the control over the entire experimental period. Cholesterol levels and concentrations of corticosterone were significantly reduced with dietary supplementation of SCFP. *E. coli*, Enterohaemorrhagic *E. coli*, ESBL-producing *Enterobacteriaceae*, and *Salmonella* counts were significantly lower in the SCFP-fed group than control/probiotic groups. Significant increase in villi height and villi-height-to-crypt-depth ratio was observed in both SCFP-fed and probiotic-fed groups. On day 28, the SCFP-fed birds and those fed with probiotics exhibited a significant increase in antibody titers against Newcastle disease virus and infectious bursal disease virus. It can be concluded that *Saccharomyces cerevisiae* fermentation product and *Bacillus subtilis* probiotic could be viable alternatives to antimicrobials in poultry production considering beneficial impacts in broilers fed an antibiotic-free diet.

## Figures and Tables

**Table 1 animals-14-00866-t001:** Ingredient and nutrient composition of basal diets.

SL. No.	Ingredients (%)	Starter (1–14 d)	Grower (15–28 d)	Finisher (29–42 d)
1	Maize	57.289	59.381	62.519
2	Soyabean meal	37.247	34.035	30.003
3	Soybean oil	1.841	3.143	4.208
4	Dicalcium phosphate	1.503	1.375	1.261
5	Limestone phosphate	0.756	0.835	0.828
6	Salt	0.322	0.324	0.326
7	DL-methionine	0.314	0.260	0.231
8	L-lysine HCL	0.226	0.154	0.131
9	L-threonine	0.084	0.055	0.055
10	Toxin Binder ^1^	0.050	0.050	0.050
11	Sodium bi-carbonate	0.100	0.100	0.100
12	Bio-Choline ^2^	0.050	0.070	0.070
13	Trace mineral mixture ^3^	0.100	0.100	0.100
14	Vitamin premix ^4^	0.100	0.100	0.100
15	Antioxidant ^5^	0.010	0.010	0.010
16	Phytase ^6^	0.010	0.010	0.010
Nutrient composition				
1	Metabolizable energy (kcal/kg) ^7^	3000.00	3100.00	3200.00
2	Crude protein (%) ^8^	22.24	20.74	19.12
3	Ether extract (%) ^8^	4.37	5.73	6.81
4	Crude fiber (%) ^8^	3.72	3.66	3.54
5	Calcium (%) ^8^	0.93	0.90	0.86
6	Available phosphorus (%) ^7^	0.45	0.42	0.39
7	Digestible lysine (%) ^7^	1.22	1.09	0.98
8	Digestible methionine (%) ^7^	0.60	0.53	0.49
9	Digestible methionine + cysteine (%) ^7^	0.88	0.80	0.74
10	Digestible threonine (%) ^7^	0.77	0.70	0.65

^1^ Niltox™, Zeus Biotech Limited, Mysore, India. ^2^ BioCholine 60, Indian Herbs Specialities Pvt. Ltd., Solan, Himachal Pradesh, India, ^3^ contains zinc 4.0%, manganese 4.0%, iron 1.5%, copper 0.8%, iodine 0.4%, selenium 300 ppm, chromium 200 ppm (Zenex animal health India Pvt. Ltd., Patiya, Ahmedabad, India), ^4^ contains vitamin E 100 g, vitamin A 40,000,000 IU, vitamin D_3_ 12,000,000 IU, pantothenic acid 60 g, vitamin K 8 g, vitamin B_1_ 120 g, vitamin B_2_ 24 g, vitamin B_6_ 10 g, vitamin B_12_ 0.10 g, biotin 0.40 g, Folic acid 4 g, niacin 100 g (DSM Nutritional Products India Pvt. Ltd. Mahabubnagar, Telangana, India). ^5^ Endox, Kemin Industries, Inc., Scott Ave Des Moines, IA, USA. ^6^ quantam blue, AB Vista, Pune, India. ^7^ Calculated values (based on the Asia South feed ingredients report 2016, Evonik Pvt Ltd., Singapore). ^8^ Analyzed values (average of triplicate values).

**Table 2 animals-14-00866-t002:** Effect of probiotic (*Bacillus subtilis*) and postbiotic *(Saccharomyces cerevisiae* fermentation products) on final body weight (BW), average daily gain (ADG), average daily feed intake (ADFI) and feed conversion ratio (FCR) and liveability of broiler chickens.

Attribute	Treatment ^1^			SEM ^2^	*p*-Value
	T_1_	T_2_	T3		
ADG (g/d)					
1–14 d	30.12 ^b^	30.35 ^b^	31.99 ^a^	0.232	0.001
15–28 d	76.19	76.10	75.11	0.377	0.446
29–42 d	85.71	87.84	89.52	1.699	0.669
1–42 d	64.01	64.76	65.54	0.565	0.554
Final BW (g)	2737.33	2770.02	2802.45	23.704	0.547
ADFI (g/d)					
1–14 d	33.74 ^b^	33.76 ^b^	35.65 ^a^	0.232	0.000
15–28 d	104.89 ^a^	102.57 ^b^	100.38 ^c^	0.431	0.000
29–42 d	154.47	153.03	153.36	1.367	0.090
1–42 d	97.70	96.45	96.46	0.533	0.561
FCR (g intake/g gain)					
1–14 d	1.12	1.11	1.12	0.058	0.922
15–28 d	1.38 ^a^	1.35 ^ab^	1.34 ^b^	0.007	0.059
29–42 d	1.81	1.76	1.73	0.022	0.293
1–42 d	1.53 ^a^	1.49 ^ab^	1.47 ^b^	0.008	0.015
Livability (%)	97.22	97.22	97.22	0.813	1.000

^abc^ Means bearing different superscripts in the same row differ significantly (*p* ≤ 0.05).^1^ The control diet (T_1_), control diet was supplemented with probiotic (*Bacillus subtilis*) at 200 mg/MT feed (T_2_), postbiotic (*Saccharomyces cerevisiae* fermentation products) at 1.25 kg/MT feed (T_3_). ^2^ SEM, standard error of means (n = 12).

**Table 3 animals-14-00866-t003:** Effect of probiotic (*Bacillus subtilis*) and postbiotic *(Saccharomyces cerevisiae* fermentation products) on carcass traits in broiler chickens at day 42.

Carcass Traits	Treatment ^1^			SEM ^2^	*p*-Value
	T_1_	T_2_	T_3_		
Slaughter body weight (g)	2718.92	2757.08	2784.58	24.096	0.549
Eviscerated carcass weight (g)	1829.83	1867.58	1874.92	16.685	0.510
Dressing Percentage	67.34	67.72	67.37	0.251	0.802
Breast (g)	731.58	739.75	754.83	12.849	0.766
Frame (g)	315.00	334.25	325.92	5.548	0.375
Thigh (g)	279.33	283.00	284.33	6.226	0.947
Drumstick (g)	254.83	272.33	268.50	5.832	0.449
Wing (g)	143.67	145.83	151.92	3.707	0.655
Neck (g)	74.71	78.83	80.21	2.204	0.584
Gizzard (g)	57.56	56.21	54.98	0.775	0.408
Liver (g)	44.92	42.20	43.43	0.702	0.294
Heart (g)	11.56	11.46	11.18	0.152	0.581
Spleen (g)	2.78	2.74	2.62	0.078	0.712
Bursa (g)	1.52	1.46	1.48	0.055	0.894
Abdominal fat (g)	38.58	40.44	42.29	0.887	0.237

^1^ The control diet (T_1_), control diet was supplemented with probiotic (*Bacillus subtilis*) at 200 mg/MT feed (T_2_), postbiotic (*Saccharomyces cerevisiae* fermentation products) at 1.25 kg/MT feed (T_3_). ^2^ SEM, standard error of means (n = 12).

**Table 4 animals-14-00866-t004:** Effect of probiotic (*Bacillus subtilis*) and postbiotic *(Saccharomyces cerevisiae* fermentation products) on blood biochemical profile (35 d), serum cortisol concentration in broiler chickens.

Attribute	Treatment ^1^			SEM ^2^	*p*-Value
	T_1_	T_2_	T_3_		
Glucose (mg/dL)	137.03	136.49	138.38	2.991	0.967
Total Protein (mg/dL)	2.86	2.88	2.67	0.060	0.309
Albumin (mg/dL)	1.77	1.79	1.72	0.065	0.912
Cholesterol (mg/dL)	118.09 ^a^	120.41 ^a^	90.01 ^b^	3.868	0.001
Triglyceride (mg/dL)	145.97	142.68	139.55	3.311	0.742
Uric Acid (mg/dL)	2.85	3.30	3.17	0101	0.181
Corticosterone(nmol/L)					
28 d	2.615	2.837	2.200	0.128	0.117
35 d	2.027 ^a^	1.840 ^a^	1.049 ^b^	0.122	0.001

^ab^ Means bearing different superscripts in the same row differ significantly (*p* ≤ 0.05). ^1^ The control diet (T_1_), control diet was supplemented with probiotic (*Bacillus subtilis*) at 200 mg/MT feed (T_2_), postbiotic (*Saccharomyces cerevisiae* fermentation products) at 1.25 kg/MT feed (T_3_). ^2^ SEM, standard error of means (n = 12).

**Table 5 animals-14-00866-t005:** Effect of probiotic (*Bacillus subtilis*) and postbiotic *(Saccharomyces cerevisiae* fermentation products) on blood hematological profile in broiler chickens on day 35.

Attribute	Treatment ^1^			SEM ^2^	*p*-Value
	T_1_	T_2_	T_3_		
Haemoglobin (g/dL)	13.45	13.72	13.85	0.520	0.953
Total leukocyte count (n × 10^3^/μL)	22.28	21.82	21.85	0.367	0.859
Heterophil (%)	33.80	33.56	32.34	0.681	0.657
Eosinophil (%)	1.80	1.55	1.58	0.228	0.890
Basophil (%)	1.91	1.24	1.65	0.235	0.516
Lymphocyte (%)	59.28	60.56	60.98	0.738	0.632
Monocyte (%)	3.22	3.09	3.46	0.268	0.857
Heterophil:lymphocyte	0.58	0.56	0.54	0.017	0.598

^1^ The control diet (T_1_), control diet was supplemented with probiotic (*Bacillus subtilis*) at 200 mg/MT feed (T_2_), postbiotic (*Saccharomyces cerevisiae* fermentation products) at 1.25 kg/MT feed (T_3_). ^2^ SEM, standard error of means (n = 12).

**Table 6 animals-14-00866-t006:** Effect of probiotic (*Bacillus subtilis*) and postbiotic *(Saccharomyces cerevisiae* fermentation products) on viable bacteria numbers (log_10_ CFU/g) in pre-cecal content in broiler chickens on day 42.

Attribute	Treatment ^1^			SEM ^2^	*p*-Value
	T_1_	T_2_	T_3_		
*Lactobacillus*	5.898	5.928	5.890	0.008	0.108
Total *E. coli*	7.377 ^a^	7.136 ^b^	7.058 ^b^	0.051	0.024
*Enterohaemorrhagic E. coli*	3.882 ^a^	3.245 ^b^	3.140 ^b^	0.661	0.000
ESBL producing *Enterobacteriaceae*	3.109 ^a^	2.833 ^b^	2.298 ^c^	0.0664	0.000
*Salmonella*	7.526 ^a^	7.045 ^b^	6.813 ^c^	0.061	0.000

^abc^ Means bearing different superscripts in the same row differ significantly (*p* ≤ 0.05).^1^ The control diet (T_1_), control diet was supplemented with probiotic (*Bacillus subtilis*) at 200 mg/MT feed (T_2_), postbiotic (*Saccharomyces cerevisiae* fermentation products) at 1.25 kg/MT feed (T_3_). ^2^ SEM, standard error of means (n = 12).

**Table 7 animals-14-00866-t007:** Effect of probiotic (*Bacillus subtilis*) and postbiotic *(Saccharomyces cerevisiae* fermentation products) on gut morphology in broiler chickens on day 42.

Attribute	Treatment ^1^			SEM ^2^	*p*-Value
	T1	T2	T3		
Duodenum					
Villi height (VH; μm)	814.33 ^b^	991.25 ^a^	1049 ^a^	37.409	0.023
Villi width (VW; μm)	92.67	89.83	84.75	3.158	0.598
Crypt depth (CD; μm)	99.83 ^a^	80.00 ^b^	79.58 ^b^	2.991	0.004
VH/CD ratio	8.33 ^b^	12.74 ^a^	13.76 ^a^	0.704	0.002
Jejunum					
Villi height (VH; μm)	818.58 ^b^	948.67 ^a^	967.75 ^a^	26.797	0.042
Villi width (VW; μm)	95.42	99.42	100.58	2.285	0.633
Crypt depth (CD; μm)	98.58 ^a^	82.75 ^ab^	78 ^b^	3.635	0.049
VH/CD ratio	8.86 ^b^	12.00 ^a^	12.68 ^a^	0.528	0.004
Ileum					
Villi height (VH; μm)	857.75 ^b^	967.08 ^ab^	1035.50 ^a^	29.147	0.036
Villi width (VW; μm)	105.67	106	106.67	3.539	0.994
Crypt depth (CD; μm)	90.83 ^a^	87.25 ^a^	74.75 ^b^	2.436	0.014
VH/CD ratio	9.65 ^b^	11.24 ^b^	14.03 ^a^	0.497	0.000

^ab^ Means bearing different superscripts in the same row differ significantly (*p* ≤ 0.05).^1^ The control diet (T_1_), control diet was supplemented with probiotic (*Bacillus subtilis*) at 200 mg/MT feed (T_2_), postbiotic (*Saccharomyces cerevisiae* fermentation products) at 1.25 kg/MT feed (T_3_). ^2^ SEM, standard error of means (n = 12).

**Table 8 animals-14-00866-t008:** Effect of probiotic (*Bacillus subtilis*) and postbiotic *(Saccharomyces cerevisiae* fermentation products) on antibody titer (log_10_) against infectious bursal disease virus (IBDV) and Newcastle disease virus (NDV), phagocytic activity of neutrophil (as expressed in optical density at 450 nm) and lymphocytes (stimulation index) in broiler chickens at 35 day.

Attribute	Treatment ^1^			SEM ^2^	*p*-Value
	T_1_	T_2_	T_3_		
Antibody titre					
IBDV-28 d	2.719 ^b^	2.808 ^ab^	3.041 ^a^	0.052	0.028
IBDV-35 d	2.757	3.009	2.871	0.066	0.307
NDV-28 d	2.608 ^b^	2.985 ^a^	2.865 ^a^	0.051	0.006
NDV-35 d	2.401	2.576	2.556	0.061	0.453
In vitro phagocytic activity					
Neutrophil	0.567	0.515	0.544	0.012	0.227
Lymphocyte	1.124	1.145	1.133	0.028	0.958

^ab^ Means bearing different superscripts in the same row differ significantly (*p* ≤ 0.05).^1^ The control diet (T_1_), control diet was supplemented with probiotic (*Bacillus subtilis*) at 200 mg/MT feed (T_2_), postbiotic (*Saccharomyces cerevisiae* fermentation products) at 1.25 kg/MT feed (T_3_). ^2^ SEM, standard error of means (n = 12).

## Data Availability

Most of the data is publicly available through the manuscript. Additional data are available on request from the corresponding author.

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
