# Peer review of "Efficacy of Saccharomyces cerevisiae Fermentation Product and Probiotic Supplementation on Growth Performance, Gut Microflora and Immunity of Broiler Chickens"

_animals, 2024, doi:10.3390/ani14060866_

Round 1

Reviewer 1 Report

Comments and Suggestions for Authors

Now, many countries are trying to restrict the use of antibiotics in animal feed. This study explored the effects of dietary supplementation with a probiotic (Bacillus subtilis) and a postbiotic (Saccharomyces cerevisiae fermentation product) on growth performance, carcass traits, blood haemato-biochemical profile, gut microbiome, gut morphology, and immune response in broilers as an alternative to antimicrobials in poultry production system. The results found that both of supplemented diets could be an alternatives to antimicrobials in broiler production. Although the innovation is slightly less prominent, the data is detailed and the workload is large, and the following problems need to be modified and improved.

1.      The abstract and conclusion are too long and cumbersome and need to be refined and condensed

2.      Why is the synergistic or antagonistic effect of the two additives (Bacillus subtilis and SCFP ) not studied? It would be even more comprehensive if a joint action group were added.

3.      The effect of dietary supplementations (T2 and T3 group) on production performance was small, but the effect on intestinal pathogenic microorganisms and non-specific and specific immunity was large, so the discussion should therefore focus on the effects on gut health and immunity.

4.      In 3.3, the results of Corticosterone was not described, but it is an important results which can explain the anti-stress effect and can be linked with the immunity effects.

5.      Line 253 to 254  and line 275 to 276 should be emphasized for the treatment groups (T3 and T2) rather than the control group (T1)

Author Response

  1. The abstract and conclusion are too long and cumbersome and need to be refined and condensed

      Reply: We agree with the reviewer and abstract/conclusion was refined

  1. Why is the synergistic or antagonistic effect of the two additives (Bacillus subtilis and SCFP ) not studied? It would be even more comprehensive if a joint action group were added.

        Reply: Synergistic/antagonistic effects of SCFP and Bacillus subtilis in broilers was not reported earlier. However, it will be considered in our future studies

  1. The effect of dietary supplementations (T2 and T3 group) on production performance was small, but the effect on intestinal pathogenic microorganisms and non-specific and specific immunity was large, so the discussion should therefore focus on the effects on gut health and immunity.

         Reply: We agree with the reviewer but considering the comments of other reviewers, we have tried to make a balance between production performance, effect on gut microbiome and immunity in the discussion section. However, additional explanations were added to justify the findings on immunity (line 471-472)

  1. In 3.3, the results of Corticosterone was not described, but it is an important results which can explain the anti-stress effect and can be linked with the immunity effects.

           Reply: We agree with the reviewer and following paragraph was added in the manuscript with references

During stress the hypothalamus pituitary–adrenal axis secretes corticosterone as the major hormone which can depress humoral immunity and decreased production of antibodies against sheep red blood cells (Honda et al., 2015). Reduced level of corticosterone in T3 group of birds during day 28 of the experiment can be correlated with increased antibody titre against NDV and IBDV. Further, corticosterone was found to be associated with upregulation of CCLi2 mRNA expression in splenic lymphocytes which attract active lymphocytes from the peripheral blood to the spleen. Corticosterone associated upregulation of CXCLi1 and CXCLi2 mRNA expression in peripheral lymphocytes instead attracts heterophiles from bone marrow which are mostly immature (Shini and Kaiser, 2009). The replacement of matured lymphocytes with immature heterophiles in the peripheral blood circulation was found responsible for decreased phagocytic activity. In the present study, reduced corticosterone level in T3 and T2 groups in comparison to T1 was found to be associated with increased (non-significant) phagocytic activity of the lymphocytes.      

  1. Line 253 to 254  and line 275 to 276 should be emphasized for the treatment groups (T3 and T2) rather than the control group (T1)

         Reply: We agree with the reviewer and the text is modified accordingly.

Reviewer 2 Report

Comments and Suggestions for Authors

Comments and Suggestions for Authors

This manuscript describes the impact of dietary supplementation with Bacillus subtilis and Saccharomyces cerevisiae fermentation product on growth performance, carcass traits, blood haemato-biochemical profile, gut microbiome, gut morphology, and immune response in broilers, and conclude Saccharomyces cerevisiae fermentation product and Bacillus subtilis could be viable alternatives to antimicrobials in poultry production. The technical approaches in this manuscript are appropriate and logic, however, I have several questions/comments that need to be addressed below.

Line 22:Change “ 324-day-old ” for “324 one-day-old” in the abstract.

Line 23:Delete the extra "."

Line 29: The full names of BW, ADFI, and FCR should be supplemented when they first appear in the abstract.

Line 34: Change “ P ” for “ p ”, please check and change all similar issues in this manuscript.

Line 40: A space should be added between T1 and group, and please check and change all similar issues in this manuscript.

Line 69: Define AGPs.

Line 149: How was the date of inoculation determined for both viruses? Is there a reference?

Lines 159-161: Need detailed description how mortality percentage adjusts BW, ADFI, and FCR calculations.

Line 203: How many volumes of formaldehyde are added?

Lines 218-221: The inoculation process of the two viruses should be described separately.

Lines 364-366: The different extraction methods of Saccharomyces cerevisiae do not appear to be covered in this manuscript.

Lines 397-400: Lower serum concentration of cholesterol in broilers is associated with production of eggs with low egg cholesterol level?The research object of this manuscript is broiler chickens, is it related to broiler hatching eggs?

Lines 456-457: Could not significantly higher anti-IBDV antibody titers be detected earlier in chickens fed with SCFP? This does not seem to fit with the results in Table 8.

Author Response

Line 22:Change “ 324-day-old ” for “324 one-day-old” in the abstract.

Reply: Modified

Line 23:Delete the extra "."

Reply: Modified

Line 29: The full names of BW, ADFI, and FCR should be supplemented when they first appear in the abstract.

Reply: Modified

Line 34: Change “ P ” for “ p ”, please check and change all similar issues in this manuscript.

Reply: Modified throughout the manuscript

Line 40: A space should be added between T1 and group, and please check and change all similar issues in this manuscript.

Reply: Modified throughout the manuscript

Line 69: Define AGPs.

Reply: Full form of AGP added

Line 149: How was the date of inoculation determined for both viruses? Is there a reference?

Reply: Reference added for common vaccination schedule of broilers followed in the country

Lines 159-161: Need detailed description how mortality percentage adjusts BW, ADFI, and FCR calculations.

Reply: It was conducted following the method described earlier (Nusairat et al., 2022).

Ref: Nusairat, B., Odetallah, N. and Wang, J.J., 2022. Live Performance and Microbial Load Modulation of Broilers Fed a Direct-Fed Microbials (DFM) and Xylanase Combination. Veterinary Sciences9(3), p.142.

Line 203: How many volumes of formaldehyde are added?

Reply: 100 ml of formaldehyde was added in 1 liter sterile distilled water

Lines 218-221: The inoculation process of the two viruses should be described separately.

Reply: Text was modified accordingly

Lines 364-366: The different extraction methods of Saccharomyces cerevisiae do not appear to be covered in this manuscript.

Reply: We agree with the reviewer that the work was conducted with the fermentation product of Saccharomyces cerevisiae and different extraction process of the yeast was not compared. We have omitted the sentence. 

Lines 397-400: Lower serum concentration of cholesterol in broilers is associated with production of eggs with low egg cholesterol level?The research object of this manuscript is broiler chickens, is it related to broiler hatching eggs?

Reply: We provide sincere thanks to the reviewer for the comment. Relationship between blood and egg cholesterol level in layers was studied earlier (Invernizzi et al., 2013) and the reference is mentioned in the manuscript. However, it is true that our study considered broilers only but it can be conducted in layers in future and the text is modified accordingly.

Lines 456-457: Could not significantly higher anti-IBDV antibody titers be detected earlier in chickens fed with SCFP? This does not seem to fit with the results in Table 8.

Reply: significantly higher antibody titre against IBDV in the birds fed with SCFP was not detected earlier as all the studies with SCFP focused NDV, not IBDV. In India, both the ND and IBD are considered as major viral infections producing constraints in optimum production (Patel et al., 2016) for which the study objective took an inclusive approach to consider both.

Ref: Patel, A.K., Pandey, V.C. and Pal, J.K., 2016. Evidence of genetic drift and reassortment in infectious bursal disease virus and emergence of outbreaks in poultry farms in India. Virus disease27(2), pp.161-169.

Reviewer 3 Report

Comments and Suggestions for Authors

please check the following lines: 48, 80, 95, 96, 117, 129, 182, 429,

also, page 3 ingredient lists #2 

This study would have been better if a fourth treatment would have been added containing the ingredients of trt 2 and trt 3. 

Comments on the Quality of English Language

please check the following lines: 48, 80, 95, 96, 117, 129, 182, 429,

also, page 3 ingredient lists #2 

Author Response

please check the following lines: 48, 80, 95, 96, 117, 129, 182, 429,

 Line 48. Deleted

Line 80. Space provided

Line 95-96. Space provided

Line 117 (119). Space provided

Line 129. Space provided

Line 182. Space provided

Line 429. Space provided

also, page 3 ingredient lists #2 ….. Space provided

 This study would have been better if a fourth treatment would have been added containing the ingredients of trt 2 and trt 3. 

Reply: Synergistic/antagonistic effects of SCFP and Bacillus subtilis in broilers was not reported earlier. However, it will be considered in our future studies